# Homeostatic regulation of perisynaptic matrix metalloproteinase 9 (MMP9) activity in the amblyopic visual cortex

**Sachiko Murase[1,2]\*, Dan Winkowski[1,2], Ji Liu[1,2], Patrick O Kanold[1,2], Elizabeth M Quinlan[1,2]\***

[1]Department of Biology, University of Maryland, College Park, United States; [2]Neuroscience Cognitive Sciences Program, University of Maryland, College Park, United States

**Abstract** Dark exposure (DE) followed by light reintroduction (LRx) reactivates robust synaptic plasticity in adult mouse primary visual cortex (V1), which allows subsequent recovery from amblyopia. Previously we showed that perisynaptic proteolysis by MMP9 mediates the enhancement of plasticity by LRx in binocular adult mice (Murase et al., 2017). However, it was unknown if a visual system compromised by amblyopia could engage this pathway. Here we show that LRx to adult amblyopic mice induces perisynaptic MMP2/9 activity and extracellular matrix (ECM) degradation in deprived and non-deprived V1. Indeed, LRx restricted to the amblyopic eye is sufficient to induce robust MMP2/9 activity at thalamo-cortical synapses and ECM degradation in deprived V1. Two-photon live imaging demonstrates that the history of visual experience regulates MMP2/9 activity in V1, and that DE lowers the threshold for the proteinase activation. The homeostatic reduction of the MMP2/9 activation threshold by DE enables visual input from the amblyopic pathway to trigger robust perisynaptic proteolysis.

**\*For correspondence:**
smurase@umd.edu (SM);
equinlan@umd.edu (EMQ)

**Competing interests:** The authors declare that no competing interests exist.

## Introduction

An imbalance in the quality of visual inputs between the two eyes during development induces amblyopia, a developmental disorder affecting up to 4% of the world's population (*Levi et al., 2015*). In animal models, prolonged monocular deprivation induces severe amblyopia, characterized by a significant decrease in the strength and selectivity of neuronal responses in the deprived visual cortex (*Fong et al., 2016*; *Harwerth et al., 1983*; *Montey et al., 2013*) and a significant loss of spatial acuity through the deprived eye (*Harwerth et al., 1983*; *Liao et al., 2011*; *Montey et al., 2013*). In rats, spatial acuity in the deprived eye is undetectable following chronic monocular deprivation (cMD) initiated at eye opening (*Eaton et al., 2016*). Additionally, following prolonged monocular deprivation, neurons in the dorsal lateral geniculate nucleus (dLGN) that project to deprived binocular visual cortex have lower metabolism (*Kennedy et al., 1981*) and smaller somata (*Duffy et al., 2018*). cMD also significantly decreases the density of dendritic spines on pyramidal neurons in deprived binocular primary visual cortex (V1b) (*Montey and Quinlan, 2011*) and induces a 60% decrease in the thalamic component of the visually evoked potential (VEP, [*Montey and Quinlan, 2011*]).

The loss of synaptic plasticity in the primary visual cortex with age is thought to significantly impede the reversal of amblyopic deficits (*Tailor et al., 2017*). Strong evidence demonstrates that developmental changes in the induction of synaptic plasticity in V1 are regulated by the maturation of fast-spiking basket interneurons (INs) that express the $Ca^{2+}$ binding protein parvalbumin (PV) (*Gu et al., 2016*; *Gu et al., 2013*; *Morishita et al., 2015*; *Stephany et al., 2016*; *Sun et al., 2016*). PV[+] INs mediate the perisomatic inhibition of pyramidal neurons, thereby exerting powerful control

of neuronal excitability and spike-timing dependent synaptic plasticity. Developmental changes in synaptic plasticity are also associated with the composition and density of the ECM, comprised of chondroitin sulfate proteoglycans (CSPGs) and hyaluronic acid, linked via cartilage link protein and tenascins (*Carulli et al., 2010*; *Celio and Chiquet-Ehrismann, 1993*; *Morawski et al., 2014*). Importantly, ECM molecules condense into perineuronal nets (PNNs) around a subset of PV+ INs, limiting their structural and functional plasticity by imposing a physical constraint and providing binding sites for molecules that inhibit neurite outgrowth (*Dickendesher et al., 2012*; *Frantz et al., 2016*; *Stephany et al., 2016*; *Vo et al., 2013*). PNNs also accumulate molecules that regulate PV+ IN excitability and maturation (*Beurdeley et al., 2012*; *Chang et al., 2010*; *Hou et al., 2017*).

Manipulations that reduce the integrity of the ECM/PNNs have been repeatedly shown to enhance plasticity in V1 and elsewhere. Treatment of adult V1 with the bacterial enzyme chondroitinase ABC (ChABC), which cleaves the CS side chains from the CSPGs, reactivates robust plasticity in rats (*Pizzorusso et al., 2006*; *Pizzorusso et al., 2002*). However, recovery from MD is incomplete in cats that receive ChABC following eye opening (*Vorobyov et al., 2013*). Enhancement of synaptic plasticity following ChABC treatment has also been demonstrated in many other brain regions (*Carstens et al., 2016*; *Carstens and Dudek, 2019*; *Gogolla et al., 2009*; *Kochlamazashvili et al., 2010*; *Romberg et al., 2013*; *Zhao et al., 2007*). Genetic ablation of cartilage link protein (Crtl1/ Halpn1), which prevents the condensation of ECM molecules into PNNs, prevents the closure of the critical period for ocular dominance plasticity (*Carulli et al., 2010*). Dark rearing from birth produces a parallel delay in the maturation of ECM/PNNs and the closure of the critical period (*Lander et al., 1997*; *Mower, 1991*; *Pizzorusso et al., 2002*).

However, robust juvenile-like synaptic plasticity can be restored in adults by complete visual deprivation through DE followed by LRx (*He et al., 2006*). Our previous work demonstrated DE/LRx induces an increase in perisynaptic activity of MMP9 and subsequent proteolysis of extracellular targets in binocular adult mice (*Murase et al., 2017*). The proteinase activity induced by LRx is perisynaptic and enriched at thalamo-cortical synapses.Importantly, the reactivation of structural and functional plasticity by DE/LRx is inhibited by pharmacological blockade and genetic ablation of MMP9. Although *Mmp9*−/− mice are resistant to enhancement of plasticity by DE/LRx, treatment with hyaluronidase activates structural and functional plasticity in adults.

In adult rats rendered severely amblyopic by cMD from eye opening to adulthood, DE followed by reverse occlusion enables recovery of the VEP amplitude and dendritic spine density in deprived V1b (*He et al., 2007*; *Montey and Quinlan, 2011*). Subsequent visual training promotes a full recovery of visual acuity in the deprived eye (*Eaton et al., 2016*). A similar reactivation of plasticity by DE has been reported in several species (*Duffy and Mitchell, 2013*; *Stodieck et al., 2014*). However, it is not known if the reactivation of plasticity in the amblyopic cortex is dependent on MMP9 activity or if this pathway can be engaged by the severely compromised visual system of an amblyope. Here we use a biomarker that reports the activity of MMP2/9 in vivo to examine the effects of DE/LRx on extracellular proteolysis in amblyopic mice. We show that DE lowers the threshold for activation of MMP2/9 by light, such that LRx to the deprived eye is sufficient to induce perisynaptic proteolysis at thalamo-cortical synapses and ECM degradation in deprived visual cortex.

## Results

### LRx activates perisynaptic MMP2/9 activity at thalamo-cortical synapses in deprived and non-deprived V1b

To test the hypothesis that LRx to amblyopic mice induces an increase in perisynaptic proteinase activity in V1b, we employed an exogenous MMP2/9 substrate in which fluorescence emission is quenched by intramolecular FRET when the substrate is intact (DQ gelatin; D12054; excitation/emission = 495/519 nm). Proteolysis of the substrate interrupts FRET and allows fluorescence emission, thereby reporting enzymatic activity. MMP9 has highly overlapping substrate specificity with MMP2 (*Szklarczyk et al., 2002*), and therefore the exogenous substrate(hereafter called biomarker) reports the activity of both metalloproteinases. However, our previous work demonstrated that LRx does not induce an increase in biomarker expression in *Mmp9*−/− mice (*Murase et al., 2017*). The MMP2/9

biomarker was delivered to V1b in vivo 24 hr prior to the onset of LRx (2 mg/ml, i.c. via cannula implanted 3 weeks prior to injection; 4 µl at 100 nl/min) and fluorescence emissionwas quantified in layer 4 of V1b 4 hr after LRx. Ex vivo imaging revealed punctate MMP2/9 activity in the deprived and non-deprived V1b (*Figure 1A*) that was similar in size, density and fluorescence intensity as previously described in binocular adult mice (*Murase et al., 2017*). No differences were observed following cMD between deprived and non-deprived V1b biomarker puncta size (deprived: $0.77 \pm 0.06$ µm$^2$, non-deprived: $0.83 \pm 0.06$ µm$^2$), density (deprived: $22.9 \pm 2.5$ puncta/0.01 mm$^2$, non-deprived: $28.9 \pm 6.6$ puncta/0.01 mm$^2$) or intensity (deprived: $42.2 \pm 2.8$ pixel, non-deprived: $40.6 \pm 2.8$ pixel, n = 6 for deprived, n = 5 for non-deprived, *Figure 1A*). However, LRx induced a significant and parallel increase in MMP2/9 biomarker puncta density (deprived: $67.0 \pm 8.0$ puncta/0.01 mm$^2$, non-deprived: $70.2 \pm 14.7$ puncta/0.01 mm$^2$) and puncta intensity (deprived: $55.6 \pm 2.6$ pixel, non-deprived: $54.6 \pm 2.8$ pixel) in deprived and non-deprived V1b, with no difference in puncta size (deprived: $0.88 \pm 0.06$ µm$^2$, non-deprived: $0.86 \pm 0.06$ µm$^2$, n = 6 subjects, One-way ANOVAs, density $F_{(3, 19)}=6.7$, p=0.003; intensity $F_{(3, 19)}=8.4$, p=0.0009; size $F_{(3, 19)}=0.52$, p=0.67; *p<0.05, Tukey-Kramer *post hoc* test; *Figure 1A*).

Biomarker puncta co-localization with thalamic axons, labelled with anti-vesicular glutamate transporter 2 (VGluT2) antibody, was low in deprived and non-deprived V1b of amblyopes (~30%), as previously described in binocular adults. LRx induced a significant and parallel increase in biomarker co-localization with VGluT2 in deprived and non-deprived V1b, indicating an increase in MMP2/9 activity at thalamo-cortical synapses (deprived: $161 \pm 6\%$, non-deprived: $199 \pm 17\%$, n = 6, 5, 6, 6 subjects for cMD dep, cMD non, LRx dep, LRx non, respectively; One-way ANOVA, $F_{(3, 19)}=9.5$, p=0.0005; *p<0.05, Tukey-Kramer *post hoc* test; *Figure 1B*). To control for false positives, we re-analyzed the co-localization following a 2 µm shift of the biomarker image relative to VGlut2. Following this manipulation we observe low co-localization of the two fluorescent signals that was unchanged by LRx (cMD dep: $7.8 \pm 3.0\%$, cMD non-dep: $5.3 \pm 2.7\%$, LRx dep: $6.9 \pm 3.0\%$, LRx non-dep: $8.4 \pm 3.0\%$) which differs significantly from co-localization of the 2 images (cMD dep: p=0.0029, cMD non-dep: p=0.036, LRx dep: $p=1.5\times10^{-5}$, LRx non-dep: $p=9.5\times10^{-6}$, paired Student's T-Test, *Figure 1B*). The LRx-induced increase in co-localization of biomarker and VGluT2 was observed at PV$^+$ neuronal somata and in PV$^-$ locations, suggesting widespread activation of perisynaptic MMP2/9 (dep: $625.3 \pm 50.1\%$, non: $484.1 \pm 35.2\%$ for PV$^+$, dep: $361.7 \pm 34.5\%$, non: $537.1 \pm 85.4\%$ for PV$^-$, n = 4 subjects; One-way ANOVA, PV$^+$ $F_{(3, 12)}=17.33$, p=0.00012; PV$^-$ $F_{(3, 12)}=56.17$, p<0.0001; *p<0.05, Tukey-Kramer *post hoc* test; *Figure 1C*).

To ask if the increase in MMP biomarker fluorescence reflects an increase in the activation of MMP9, we performed quantitative western blots for the active MMP9 isoform (95 kDa), which can be distinguished from inactive pro-MMP9 (105 kDa, [*Szklarczyk et al., 2002*]). Quantitative immuno-blot analysis showed that DE followed by 2 hr of LRx significantly increased the concentration of active MMP9 in parallel in deprived and non-deprived V1b of adult cMD mice (% of cMD dep: $133.04 \pm 9.1\%$; non: $131.37 \pm 8.9\%$; n = 7, eight subjects for cMD and LRx, respectively; One-way ANOVA, $F_{(3, 26)}=5.6$, p=0.004; *p<0.05, Tukey-Kramer *post hoc* test; *Figure 1D*).

## LRx induces a parallel degradation of ECM in deprived and non-deprived V1b

MMP9 has several extracellular targets, including aggrecan (Agg) the predominant CSPG in the ECM of the adult mammalian cortex (*Mercuri et al., 2000*). To test the hypothesis that LRx to the amblyopic cortex induces ECM degradation, we examined the distribution of Wisteria floribunda agglutinin (WFA), a plant lectin that binds to the CS side chains of CSPGs, combined with immunoreactivity for Agg and PV. Diffuse WFA and Agg fluorescence was observed throughout the depth of the V1b, with a peak between 250–400 µm from the cortical surface. In addition, WFA and Agg fluorescence was concentrated around the perisomatic area of a subset of PV$^+$ INs (*Figure 2A and B*, WFA and IHC of PV in *Figure 2—figure supplement 1*). Importantly, the intensity and distribution of WFA and Agg fluorescence were similar between deprived and non-deprived V1b. LRx induced a significant decrease in the intensity of WFA and Agg, but not PV labeling, 250 µm-400 µm from the cortical surface (mean ± SEM; % of cMD, WFA dep $54.3 \pm 3.5$, non $46.3 \pm 3.4$, One way ANOVAs, $F_{(3, 26)}=32$, p<0.0001; Agg dep $62.3 \pm 3.1$, non $54.3 \pm 4.5$, $F_{(3, 26)}=10$, p=0.0001; PV dep $105.0 \pm 6.9$, non $96.4 \pm 4.5$; $F_{(3, 26)}=0.4$, p=0.75; n = 7, 7, 8, 8 subjects for cMD dep, cMD non, LRx dep, LRx non, respectively; *p<0.05, Tukey-Kramer *post hoc* test; *Figure 2C*). Line scans of triple-labeled images

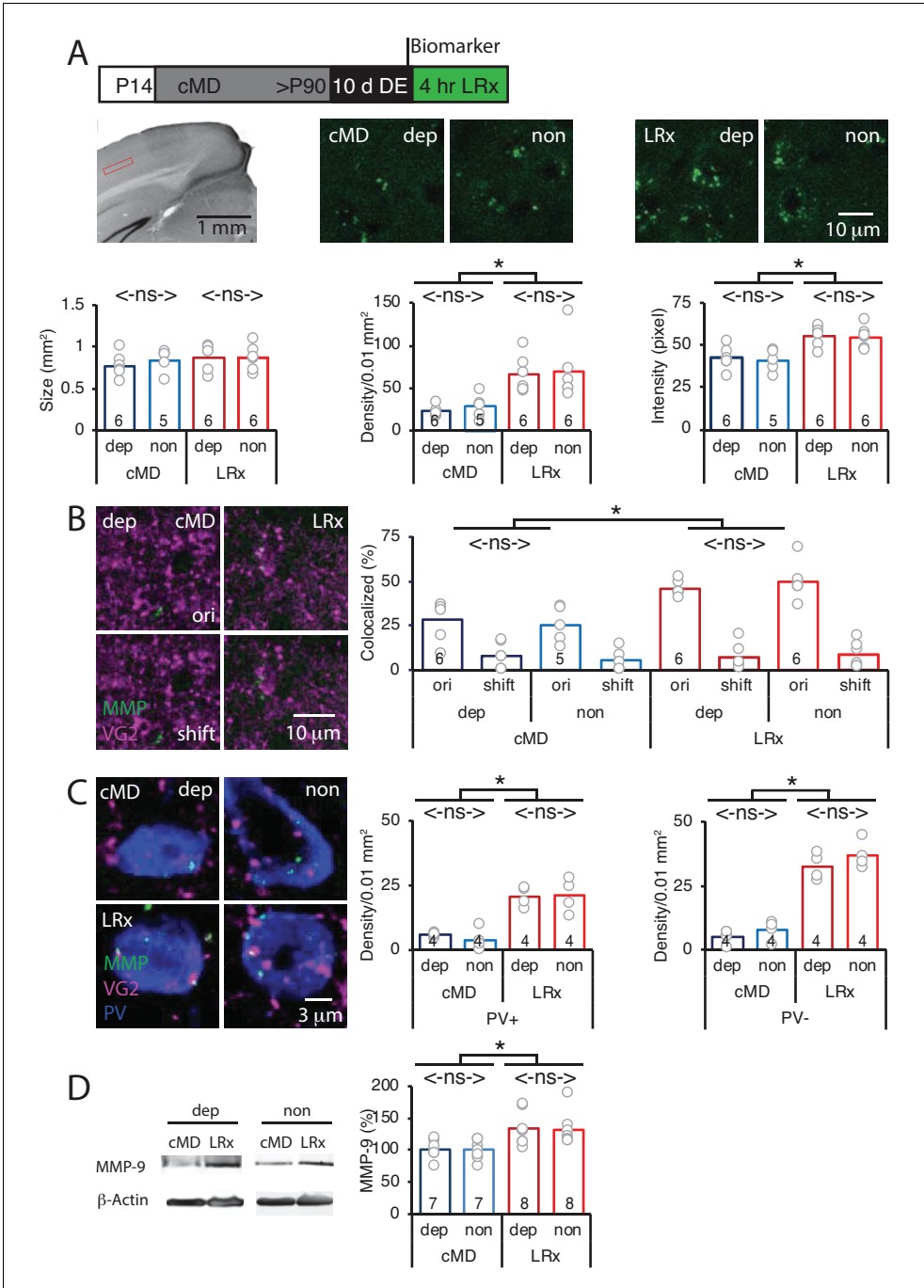

**Figure 1.** Parallel increase in MMP2/9 activity following LRx at thalamo-cortical synapses in deprived and non-deprived V1b. (**A**) Top: Experimental timeline. Subjects received cMD from eye opening (postnatal day 14, (P14) until adulthood (>P90). 10 days of DE was followed by 4 hr of LRx. MMP2/9 biomarker (4 μl of 2 mg/ml Dye-quenched gelatin) was delivered i.c. 24 hr prior to 4 hr of LRx. Middle left: Coronal section with DAPI nuclear staining. Layer 4 of binocular region of V1 indicated by red box. Middle right: representative images of MMP2/9 biomarker fluorescence in deprived (dep) and non-deprived (non) V1b in cMD (left) and cMD+LRx subjects (LRx, right). Bottom: Quantification of biomarker puncta reveals no change in puncta size (left), but a parallel and significant increase in puncta density (middle) and fluorescent intensity (right) in dep and non V1b following LRx. One-way ANOVAs, size $F_{(3, 19)}$=0.52, p=0.67; density $F_{(3, 19)}$=6.7, p=0.003; intensity $F_{(3, 19)}$=8.4, p=0.0009; n = 6, 5, 6, 6 subjects for cMD dep, cMD non, LRx dep, LRx non, respectively; *p<0.05, Tukey-Kramer *post hoc* test. (**B**) Representative images of MMP2/9 biomarker fluorescence (MMP; green) and marker for thalamic axons (VG2; magenta) in deprived visual cortex in cMD and cMD+LRx subjects. A parallel and significant increase in

*Figure 1 continued on next page*

*Figure 1 continued*

colocalization of biomarker puncta with VGluT2 following LRx in dep and non V1b. One-way ANOVA, $F_{(3, 19)}$=9.5, p=0.0005; n = 6, 5, 6, 6 subjects for cMD dep, cMD non, LRx dep, LRx non, respectively; *p<0.05, Tukey-Kramer *post hoc* test. Co-localization with VGluT2 is lost following 2 μm shift of biomarker image (shift). (C) Representative images of MMP2/9 biomarker (green), VGluT2 (magenta) and parvalbumin fluorescence (PV; blue) in deprived and non-deprived cMD and cMD+LRx subjects. Significant increase in co-localization of biomarker puncta with VGluT2 at PV$^+$ and PV$^-$ locations of dep and non V1b following LRx. One-way ANOVAs, PV$^+$ $F_{(3, 12)}$=17.33, p=0.00012; PV$^-$ $F_{(3, 12)}$=56.17, p<0.0001; n = 4, 4, 4, 4 subjects for cMD dep, cMD non, LRx dep, LRx non, respectively; *p<0.05, Tukey-Kramer *post hoc* test. (D) Left: Representative immunoblots for active MMP9 (95 kDa), and β-actin from dep and non V1b. MMP9 level is normalized to β-actin and reported as % of cMD non. Right: Quantification of immunoblots reveals a parallel and significant increase in active MMP9 in dep and non V1b following 2 hr of LRx. One-way ANOVA, $F_{(3, 26)}$=5.6, p=0.004; n = 7, 7, 8, 8 subjects for cMD dep, cMD non, LRx dep, LRx non, respectively: *p<0.05, Tukey-Kramer *post hoc* test.

The online version of this article includes the following source data for figure 1:

**Source data 1.** Source data for *Figure 1*.

revealed that the decrease in WFA and Agg staining was observed at PV$^+$ and PV$^-$ locations suggesting widespread degradation of ECM upon LRx (% of cMD, WFA PV$^+$ dep 56.3 ± 2.1, non 55.4 ± 2.1; One-way ANOVAs, $F_{(3, 309)}$=40.3, p<0.0001; WFA PV$^-$ dep 69.5 ± 1.9, non 66.8 ± 1.3; $F_{(3, 309)}$=30.1, p<0.0001; Agg PV$^+$ dep 61.9 ± 1.4, non 60.2 ± 1.4; $F_{(3, 309)}$=29.4, p<0.0001; Agg PV$^-$ dep 77.1 ± 1.2, non 71.0 ± 1.3; $F_{(3, 309)}$=18.7, p<0.0001; n (subjects, ROIs)=(5, 77 , 5, 73 , 5, 81 , 5, 82), for cMD dep, cMD non, LRx dep, LRx non, respectively; *p<0.05, Tukey-Kramer *post hoc* test; *Figure 2D*).

## LRx to cMD eye is sufficient to activate MMP2/9 at thalamic input to deprived V1

The LRx-induced increase in MMP2/9 activity and degradation of ECM in the deprived V1b could reflect activity of either deprived or non-deprived eye inputs to V1b. To ask if LRx to the amblyopic eye was sufficient to drive an increase in MMP2/9 activity in deprived V1b, LRx was delivered to adult amblyopes with a light-occluding eye patch covering the non-deprived eye. LRx restricted to the amblyopic eye induced an increase in MMP2/9 biomarker density and intensity in layer four in deprived V1b (contralateral to the cMD eye, ipsilateral to eye patch) relative to non-deprived V1b (ipsilateral to the cMD eye, contralateral to eye patch; density: 208.8 ± 23.1% of non, n = 6 subjects, p=0.014; intensity: 161.1 ± 22.9% of non, n = 6 subjects, p=0.046, Student's T-test), with no change in biomarker puncta size (111.7 ± 14.8% of non, n = 6 subjects, p=0.56, Student's T-test; *Figure 3A*). Similarly, LRx restricted to the amblyopic eye induced a significant increase in the co-localization of MMP2/9 biomarker puncta with VGluT2 in deprived relative to non-deprived V1b (dep: 57.8 ± 3.5%; non: 33.4 ± 5.0%, n = 6 subjects, p=0.0026, Student's T-test; *Figure 3B*). Co-localization with VGluT2 following a 2 μm shift of the biomarker image was low: dep: 8.0 ± 3.9%; non: 7.0 ± 3.9%, and significantly different from co-localization with the two images correctly registered (dep: p=3.2×10$^{-5}$; non: p=0.0017, paired Student's T-test; *Figure 3B*). Again, the LRx-induced increase in co-localization of MMP2/9 biomarker and VGluT2 was observed at PV$^+$ and PV$^-$ locations (PV$^+$: 334 ± 91% of non, n = 4 subjects, p=0.04, PV$^-$: 271 ± 38% of non, n = 4 subjects, p=0.03, Student's T-test; *Figure 3C*).

## LRx to deprived eye is sufficient to induce ECM degradation in deprived V1b

To ask if LRx restricted to the amblyopic eye is sufficient to induce ECM degradation in deprived V1b, we again employed a light-occluding patch on the non-deprived eye. LRx to the deprived eye alone induced a decrease in the mean fluorescence intensity of WFA and Agg in layer 4 of deprived V1b (contralateral to cMD eye, ipsilateral to eye patch) relative to non-deprived V1b (ipsilateral to cMD eye, contralateral to eye patch; quantified 250 μm - 400 μm from the cortical surface: WFA: 48.7 ± 7.0% of non, n = 5 subjects, p=0.0009; Agg: 50.2 ± 16.1% of non, n = 5 subjects, p=0.02, Student's T-test), with PV fluorescence unchanged (83.3 ± 17.7% of non-deprived, n = 5 subjects, p=0.4, Student's T-test; *Figure 4A–C*, WFA and IHC of PV in *Figure 4—figure supplement 1*). Line scans of triple-labeled images revealed that the decrease in WFA and Agg staining was observed at

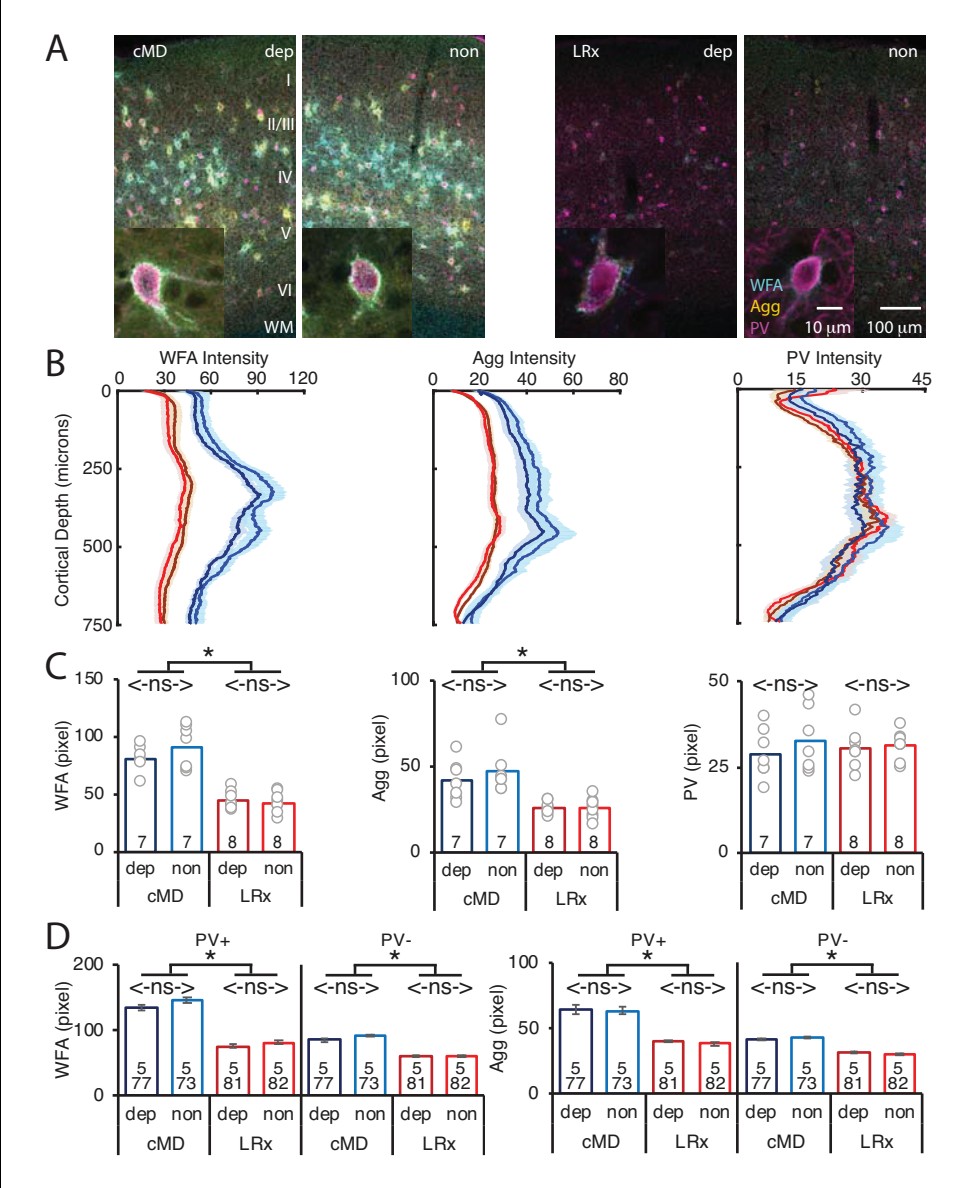

**Figure 2.** Parallel decrease in ECM integrity following LRx in deprived and non-deprived V1b. (A) Representative triple labeled fluorescent micrographs of Wisteria floribunda agglutinin (WFA)-FITC staining (cyan), immunostaining for aggrecan (Agg; yellow) and parvalbumin (PV; magenta) in deprived (dep) and non-deprived (non) V1b in cMD (left) and cMD+LRx subjects (LRx, right). Roman numerals indicate cortical layer. WM = white matter. Inset: High magnification images of triple labeled PV[+] interneurons (100X). (B) Fluorescence intensity profiles (mean ± SEM) along vertical depth of V1b. cMD dep (dark blue), cMD non (light blue), LRx dep (dark red), LRx non (red). (C) A parallel and significant decrease in WFA and Agg mean fluorescence intensity 250–400 µm from surface in dep and non V1b following LRx. One-way ANOVAs, WFA $F_{(3, 26)}$=32, p=0.0001; Agg $F_{(3, 26)}$=10, p=0.0001; PV $F_{(3, 26)}$=0.4, p=0.75; n = 7, 7, 8, eight subjects for cMD dep, cMD non, LRx dep, LRx non, respectively; *p<0.05, Tukey-Kramer *post hoc* test. (D) LRx induces a significant decrease in WFA and Agg fluorescence intensity at PV[+] and PV[-]locations in dep and non V1b. One-way ANOVAs, WFA PV[+], $F_{(3, 309)}$=40.3, p<0.0001; WFA PV[-], $F_{(3, 309)}$=30.1, p<0.0001; Agg PV[+], $F_{(3, 309)}$=29.4, p<0.0001; Agg PV[-], $F_{(3, 309)}$=18.7, p<0.0001; n (subjects, ROIs)= (5, 77 , 5, 73 , 5, 81 , 5, 82), for cMD dep, cMD non, LRx dep, LRx non, respectively; *p<0.05, Tukey-Kramer *post hoc* test.

The online version of this article includes the following source data and figure supplement(s) for figure 2:

**Source data 1.** Source data for *Figure 2*.

**Figure supplement 1.** Parallel decrease in ECM integrity following LRx in deprived and non-deprived V1.

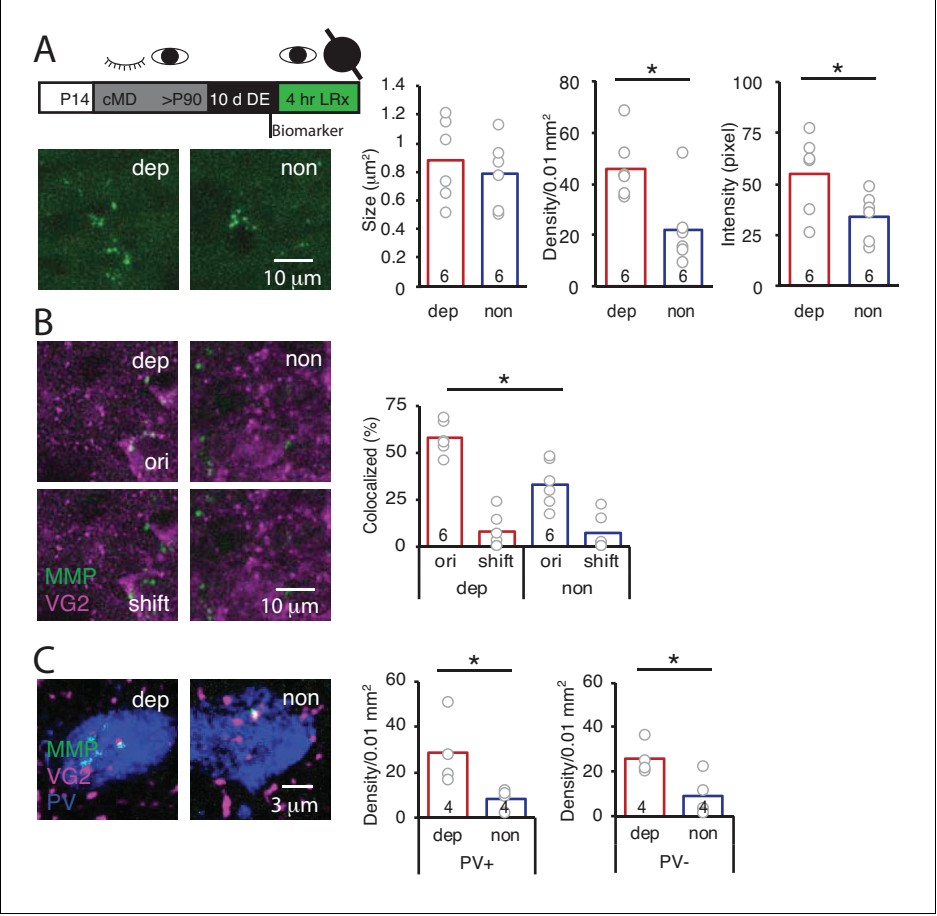

**Figure 3.** LRx limited to deprived eye is sufficient to induce perisynaptic MMP2/9 activity at thalamo-cortical synapses in deprived V1b. (**A**) Top: Experimental timeline. A light-occluding eye patch was attached to the non-deprived eye before DE. Bottom left: Representative images of MMP2/9 biomarker fluorescence in layer 4 of chronically deprived (dep, contralateral to cMD, ipsilateral to eye patch) and non-deprived (non, ipsilateral to cMD, contralateral to eye patch) V1b of LRx subject. Quantification of biomarker puncta reveals a significant increase in density and intensity in dep vs non V1b following LRx to amblyopic eye; n = 6, six subjects for LRx dep, LRx non, respectively; *p<0.05, Student's T-test. (**B**) Representative images of MMP2/9 biomarker fluorescence (MMP, green) and VGluT2 immunoreactivity (VG2, magenta) in dep and non V1b of LRx subject. A significant increase in biomarker colocalization with VGluT2 in dep vs non V1b following LRx to amblyopic eye; n = 6, six subjects for LRx dep, LRx non, respectively; *p<0.05, Student's T-test. Co-localization with VGluT2 is reduced following 2 μm shift of biomarker image (shift). (**C**) Representative images of MMP2/9 biomarker (green), VGluT2 (magenta) and parvalbumin fluorescence (PV, blue) in dep and non V1b following LRx. A significant increase in co-localization of MMP2/9 biomarker puncta with VGluT2 at PV[+] and PV[-] immunoreactive locations of dep vs non V1b; n = 4; *p<0.05, Student's T-test.

The online version of this article includes the following source data for figure 3:

**Source data 1.** Source data for *Figure 3*.

PV[+] and PV[-] locations (% of non: WFA PV[+]: 40.1 ± 2.4%, WFA PV[-]38.7 ± 1.0%, Agg PV[+]: 37.0 ± 1.9%, Agg PV[-]: 52.4 ± 2.0% of non; all p<0.001, Student's T-test; n (subjects, ROIs) = (5, 73 , 5, 78), for LRx dep, LRx non, respectively; *Figure 4D*).

## DE regulates threshold for perisynaptic MMP2/9 activation

Our previous work demonstrated that MMP2/9 biomarker expression is low in the adult visual cortex, indicating that ambient light does not typically stimulate activatioin of this proteinase (*Murase et al., 2017*). The robust induction of MMP2/9 activity by reintroduction to ambient light, therefore, suggests that DE may lower the threshold for MMP2/9 activation. To test this prediction,

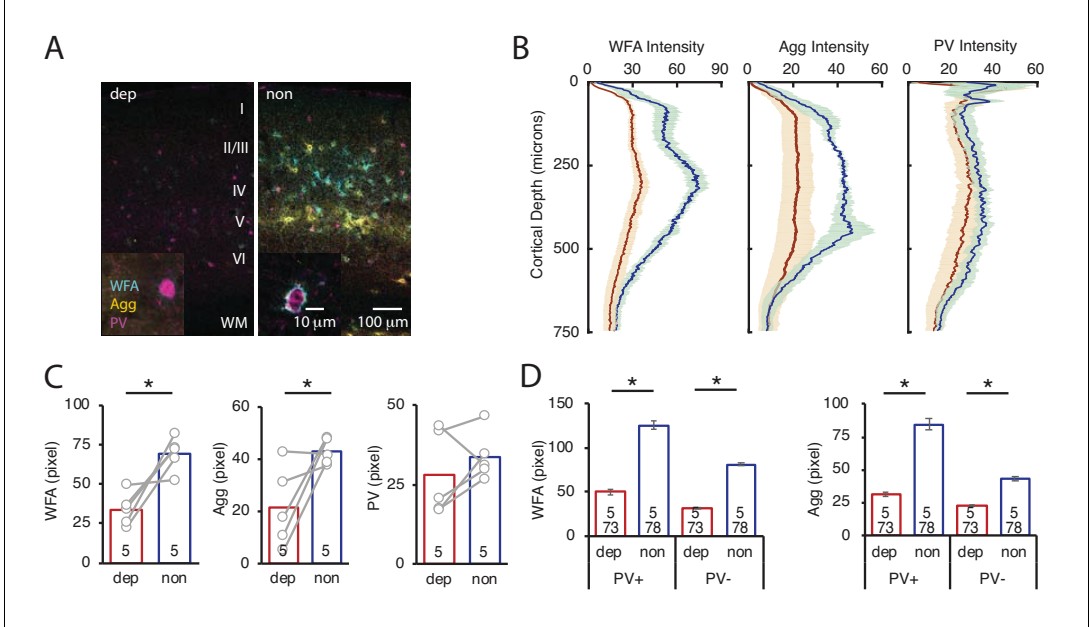

**Figure 4.** LRx limited to deprived eye is sufficient to decrease ECM integrity in deprived V1b. (A) Representative triple labeled fluorescent micrographs of WFA-FITC staining (cyan), immunostaining for aggrecan (Agg; yellow) and parvalbumin (PV; magenta) of deprived (dep, contralateral to cMD, ipsilateral to eye patch) and non-deprived (non, ipsilateral to cMD, contralateral to eye patch) V1b after LRx to amblyopic eye. Roman numerals indicate cortical layer. WM = white matter. Inset: High magnification images of triple labeled PV[+] interneurons (100X). (B) Fluorescence intensity profiles (mean ± SEM) along vertical depth of V1b. Dep LRx (dark red), non LRx (blue). (C) A significant decrease in WFA and Agg mean fluorescence intensity 250–400 µm from surface in dep V1b; n = 5; *p<0.05, Student's T-test. (D) LRx-induced a significant decrease in WFA and Agg fluorescence intensity at PV[+] and PV[-] locations in dep V1b; n (subjects, ROIs)=(5, 73 , 5 78), for LRx dep, LRx non, respectively; *p<0.05, Student's T-test.
The online version of this article includes the following source data and figure supplement(s) for figure 4:

**Source data 1.** Source data for *Figure 4*.
**Figure supplement 1.** LRx limited to deprived eye is sufficient to decrease ECM integrity in deprived V1b.

we used two-photon live imaging of the MMP2/9 biomarker in awake binocular mice. In these experiments, visual deprivation must be maintained during i.c. delivery of the MMP2/9 biomarker and during two-photon imaging following DE. To achieve this, we designed a novel imaging chamber containing an aperture for the placement of the objective on the cranial imaging window without light exposure (*Figure 5A*). In addition, we employed a small peptide MMP2/9 biomarker (A580, Mw: 1769, Anaspec) that diffuses ~1 cm from the injection site, which allowed delivery at the cranial window margin. The biomarker is a synthetic substrate for MMP2/9, containing a C-terminal fluorescent donor carboxy-tetramethyl-rhodamine (TAMRA) and an N-terminal non-fluorescent acceptor QXL (quencher) for intramolecular FRET. In the absence of MMP2/9 activity intramolecular FRET (from TAMRA to QXL) quenches fluorescence emission following excitation at 547 nm. If the peptide is cleaved by MMP2/9, intramolecular FRET is interrupted. We cfirst onfirmed the fluorescence emission peak (Em max = 585 nm) following excitation at 545 nm (*Figure 5B*) in vitro, demonstrating that the biomarker reports the activation of recombinant MMP. In vivo 2P live imaging of biomarker was acquired simultaneously with GFP in pyramidal neurons following AAV-CAM-KII-GFP delivery. Adult mice received DE for 10 d prior to 40 s of light stimulation (470 nm LED, 1 Hz flash). Dual imaging of biomarker and GFP revealed that both signals remained stable in the absence of visual stimulation (light intensity = 0 cd/m$^2$). In contrast, stimulation with moderate intensity light (300 cd/m$^2$; equivalent to luminance in the laboratory) induced an increase in raw fluorescence and biomarker ΔF/F of the biomarker, but not co-localized GFP (*Figure 5C*). Population data reveals a significant increase in biomarker fluorescence in DE subjects following moderate intensity light stimulation (repeated measure ANOVA, $F_{(1, 22)}$, p<0.001; n = 12 puncta from three subjects; *p<0.001, Tukey-Kramer *post hoc* test; *Figure 5D*). The change in biomarker fluorescence was similar to that observed following higher intensity light stimulation (150,000 cd/m$^2$, equivalent to direct

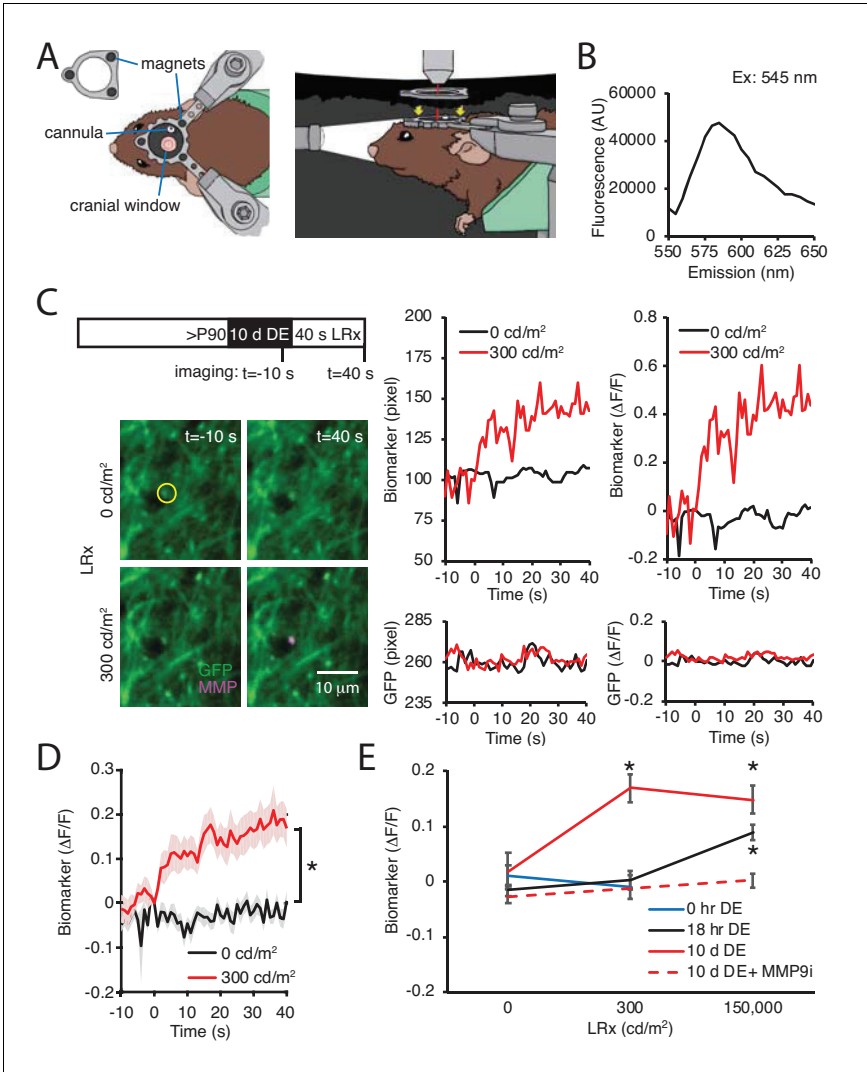

**Figure 5.** DE lowers the threshold for light-induced activation of MMP2/9. (A) Dark chamber with an imaging window allows maintenance of visual deprivation during two photon live imaging of MMP2/9 biomarker. Left drawing: Top view of a subject wearing a custom aluminum headpost (1 cm diameter) magnetically held to an o-ring in the blackout ceiling of the dark camber (inset; 3 mm diameter magnets APEX magnets; magnetic field generation around V1,<20 gauss). The headpost is secured to a stereotax. The cannula for biomarker delivery is adjacent to theimaging window margin. Right drawing: Side view of a subject in the dark chamber. The headpost is magnetically attached to the o-ring opening of the blackout ceiling (magnet locations, yellow arrows). (B) In vitro emission spectrum of MMP2/9 biomarker A580 (2 ng/ml) incubated with activated rat recombinant MMP9 (rrMMP9, 100 ng). (C) Inset: Experimental timeline. Adult (>P90) WT mice received AAV-CaMKII-GFP~2 weeks before 10 d of DE. Biomarker was delivered 24 hr before imaging. Subjects received 40 s of light stimulation (1 Hz flash of 470 nm LED at 0 or 300 cd/m$^2$). Left: Representative images of GFP (green) and biomarker (MMP, magenta) signals in V1b 10 s prior or 40 s after light stimulation at 0 or 300 cd/m$^2$ in a DE subject. Right: Time course of raw fluorescent intensities (pixel) and ΔF/F of MMP biomarker (top) and co-localized GFP (bottom) within the single ROI denoted by yellow circle, from 10 s before (−10) to 40 s after (+40) light stimulation of 0 or 300 cd/m$^2$ in a DE subject. (D) Summary data: Time course of ΔF/F of MMP biomarker from −10 s to +40 s of light stimulation of 0 or 300 cd/m$^2$ in DE subjects. ΔF/F of MMP biomarker was stable in absence of visual stimulation (0 cd/m$^2$) and increased over time in response to 300 cd/m$^2$ light stimulation (mean ± SEM; Repeated measure ANOVA, $F_{(1, 22)}$, *p<0.001; n = 12 puncta from three subjects each). (E) Biomarker ΔF/F +40 s relative to 0 s as a function of DE (0, hr, 18 hr or 10 d) and light intensity (0, 300, or 150,000 cd/m$^2$). Moderate intensity light did not induce a change in biomarker fluorescence in the absence of DE (blue line, p=0.49, Student's T-test; n = 11, 10 puncta for 0 and 300 cd/m$^2$, respectively). Following 18 hr of dark adaptation, a significant increase in biomarker fluorescence was observed in response to high, but not moderate intensity light (black line, One-way ANOVA, $F_{(2,}$

*Figure 5 continued on next page*

*Figure 5 continued*

$_{45)}$=11.5, p<0.0001; n = 12, 12, 24 puncta for 0, 300, 150,000 cd/m$^2$, respectively, *p<0.05, Tukey-Kramer *post hoc* test). Following 10 d DE, moderate and high intensity light significantly increased biomarker fluorescence (solid red line, One-way ANOVA, F$_{(2, 40)}$=6.3, p=0.0042; n = 12, 12, 19 puncta for 0, 300, 150,000 cd/m$^2$, respectively, *p<0.05, Tukey-Kramer *post hoc* test). The increase in biomarker fluorescence by 150,000 cd/m$^2$ stimulation to 10 d DE subjects was inhibited by an MMP9 inhibitor delivered 24 hr before visual stimulation (MMP9i; 5 nM delivered i.c. 24 hr prior to LRx, dashed red line, p=0.13 Student's T-test; n = 13 puncta for 0 and 150,000 cd/m$^2$). The online version of this article includes the following source data for figure 5:

**Source data 1.** Source data for *Figure 5*.

sunlight at noon; One-way ANOVA, F$_{(2, 40)}$=6.3, p=0.0042, n = 12, 12, 19 puncta from three subjects each, for 0, 300, 150,000 cd/m$^2$, respectively; *Figure 5E*). Importantly, moderate intensity light stimulation (300 cd/m$^2$) was insufficient to induce an increase in biomarker fluorescence in control (0 hr DE) or dark-adapted (18 hr DE) subjects (*Figure 5E*). To confirm the specificity of the biomarker in reporting activity of MMP2/9, DE subjects were stimulated with high intensity light in the presence of a potent and specific inhibitor of MMP9 (Inhibitor I, CAS 1177749-58-4; IC$_{50}$ = 5 nM). Co-delivery of MMP9 inhibitor (MMP9i, 5 nM 4 µl at 100 nl/min) with biomarker 24 hr prior to LRx inhibited the increase in biomarker fluorescence by high intensity light (n = 13 puncta from 3 subjects for 0 and 150,000 cd/m$^2$; Student's T-test; p=0.13). Together this demonstrates that DE lowers the threshold for light-induced MMP2/9 activation in the adult visual cortex.

## Discussion

Activity-dependent induction of perisynaptic proteolysis is a powerful mechanism to couple experience with the activation of synapse-specific plasticity. Indeed, in the adult visual cortex, the threshold for activation of MMP2/9 is high and basal activity is low, favoring stability over plasticity. Brief dark-adaptation does not modify the basal activity of MMP2/9, consistent with the stability of the ECM over the light/dark cycle. In contrast, prolonged DE lowers the threshold for the induction of perisynaptic MMP2/9 activity, allowing for moderate light stimulation to trigger proteolysis in V1b. Following prolonged DE, LRx restricted to the amblyopic eye is sufficient to induce a robust and widespread increase in MMP2/9 activity, including at thalamic inputs to cortical neurons, which is blocked by a potent and specific inhibitor of MMP9. These observations, in conjunction with our previous demonstration that MMP2 activity was unchanged following LRx (*Murase et al., 2017*) indicate that the induction of MMP9-dependent perisynaptic proteolysis at thalamic inputs to cortical neurons by ambient light is enabled in the amblyopic visual by DE/LRx.

Prolonged DE induces several changes in the composition of function of synapses in the primary visual cortex that may influence the activation threshold for MMP2/9. For example, during DE, the NMDA subtype of glutamate receptor reverts to the juvenile form, characterized by the presence of high levels of GluN2B subunit and an increase in the temporal summation of NMDAR-mediated EPSCs (*He et al., 2006*; *He et al., 2007*; *Yashiro et al., 2005*). The change in NMDAR composition and function predicts that the threshold for Hebbian synaptic plasticity is lowered by DE (*Cooper and Bear, 2012*). Indeed, following 3 days of DE, spontaneous activity is sufficient to induce an GluN2B-dependent potentiation of excitatory synapse strength in V1 (*Bridi et al., 2018*). However, prolonged DE alone does not induce a change in MMP2/9 biomarker expression or ECM integrity (*Murase et al., 2017*).

It is well-appreciated that Hebbian plasticity is reduced at thalamo-cortical synapses early in postnatal development, and this loss is associated with the closure of critical periods in barrel, visual and auditory cortices (*Crair and Malenka, 1995*; *Glazewski and Fox, 1996*; *Kirkwood et al., 1995*; *Sun et al., 2018*). The sparse co-localization of MMP2/9 biomarker with VGluT2 in binocular and amblyopic adults demonstrates that baseline activity of the protease is also low at thalamo-cortical synapses. Indeed, MMP2/9 activity is 3X higher in VGluT1$^+$ cortico-cortical synapses than VGluT2$^+$ thalamo-cortical synapses in binocular adult mice reared in a normal light:dark cycle. DE alone does not increase MMP9 activity at thalamo-cortical synapses or change the amplitude of optogenetically-evoked thalamo-cortical EPSCs (*Petrus et al., 2014*). In contrast, LRx induces an increase in perisynaptic MMP2/9 activity that is highly enriched at thalamo-cortical synapses in V1b in binocular and

amblyopic adult mice. LRx stimulates a 2X increase in MMP2/9 activity at VGluT2$^+$ relative to VGluT1$^+$ synapses, consistent with different trheshold for activation at different classes of synapses. We have previously shown that the LRx induced perisynaptic proteolysis at thalamic inputs to PV$^+$ INs is coincident with a decrease in the visually-evoked activity in FS INs in binocular adults (*Murase et al., 2017*). However, the LRx-induced increase in MMP2/9 activity in PV$^+$ and PV$^-$ locations predicts enhanced plasticity at thalamic inputs to multiple classes of neurons in the adult cortex.

Several mechanisms could contribute to the activity-dependent homeostatic regulation of MMP9. MMP9 is activated following cleavage from the inactive pro-MMP9 by other proteases, including plasmin (*Davis et al., 2001*) and inhibited endogenously by tissue inhibitor of metalloproteinase 1 (TIMP1, [*Candelario-Jalil et al., 2009*]). Although TIMP1 is co-released from vesicles with MMP9 (*Sbai et al., 2008*), activity-dependent stimulation of synaptic mRNA translation (*Dziembowska et al., 2012*) could perturb the ratio of MMP9 to TIMP1 at thalamo-cortical synapses. Alternatively, the activation of MMP9 at thalamo-cortical synapses by LRx may be due to synapse-specific release of tissue activator of plasminogen (tPA, [*Lochner et al., 2006*]) or synapse-specific Ca$^{2+}$ signaling, as inhibition of CaMKII blocks depolarization-induced proteolysis by MMP9 (*Peixoto et al., 2012*). Indeed, tPA levels increase following MD in the visual cortex during critical period (*Mataga et al., 2004*). However, the tPA-induced increase in dendritic spine motility is occluded by MD only in extragranular and infragranular layers (*Oray et al., 2004*), indicating that endogenous tPA activity may be weak in thalamo-recipient granular layer.

The role of MMP9 in the promotion of synaptic plasticity has been best described in the hippocampus. LTP-inducing tetanic stimulation of CA1 neurons in slices from adult rats increases perisynaptic MMP9 activity at CA3 neuron dendritic spines (*Bozdagi et al., 2007*). Similarly, an increase in perisynaptic MMP9 activity has been shown to be coincident with the enlargement of dendritic spines induced by a chemical LTP protocol (*Szepesi et al., 2014*). MMP9 activity has also been correlated with synaptic weakening and synaptic loss in the hippocampus (*Bemben et al., 2019*; *Peixoto et al., 2016*). Interestingly, in kindling-induced epilepsy, MMP9 activity is associated with the pruning of dendritic spines and aberrant synaptogenesis after mossy fiber sprouting in hippocampus (*Wilczynski et al., 2008*).

MMP9 has been implicated in the response to MD during the critical period, in which a rapid, NMDAR-dependent depression of synapses serving the deprived eye is followed by a slow potentiation of synapses serving the non-deprived eye. Accordingly, non-deprived eye strengthening may compensate for deprived eye weakening to maintain activity levels in the deprived visual cortex. The similarity in ECM integrity in the visual cortex contralateral versus ipsilateral to the chronically deprived eye supports this idea Accordingly, non-deprived eye strengthening may compensate for deprived eye weakening to maintain activity levels in the deprived visual cortex. The similarity in ECM integrity in the visual cortex contralateral versus ipsilateral to the chronically deprived eye supports this idea. Deprived eye depression persists following short-term pharmacological inhibition of MMP9 in rats, but non-deprived eye strengthening is inhibited (*Spolidoro et al., 2012*). However, both deprived eye weakening and non-deprived eye strengthening were compromised following brief MD during the critical period in the *Mmp9$^{-/-}$* mouse (*Kelly et al., 2015*).

Our previous work demonstrates that the full recovery from cMD in adulthood requires reverse deprivation and subsequent visual training after DE (*Eaton et al., 2016*; *He et al., 2007*). However, asynchronous activity through the reverse deprived eye could depress deprived eye synapses and transfer amblyopia to the originally non-deprived eye (*Mitchell, 1991*). Here we employed a light-occluding eye patch to block the low spatial frequency vision that persists during monocular lid suture. Interestingly, we observed no change in perisynaptic MMP2/9 activity or ECM integrity in V1b contralateral to the occluding patch.

An array of homeostatic mechanisms have been described in the central nervous system that serve to maintain the range of circuit function following changes in activity patterns (*Abraham, 2008*; *Keck et al., 2017*; *Li et al., 2019*). In response to a severe reduction in activity, the strength of inhibitory synapses onto excitatory neurons decreases, while the strength of excitatory synapses on excitatory neurons and intrinsic excitability increase (*Blackman et al., 2012*; *Chang et al., 2010*; *Desai et al., 1999*; *Turrigiano et al., 1998*). In contrast, a reduction in excitatory drive lowers the threshold for Hebbian plasticity, promoting the potentiation of active synapses (*Cooper and Bear, 2012*). Here we expand the list of synaptic parameters that are regulated homeostatically by

demonstrating the lowereing of the threshold for light-induced activation of MMP2/9 following DE. In vivo live imaging of the MMP2/9 biomarker revealed that ambient light was sufficient to induce perisynaptic MMP2/9 activity only after prolonged DE. However, 18 hr of dark adaptation induced response to high intensity light. The LRx induced increase in biomarker fluorescence was blocked by a potent and specific inhibitor of MMP9.

Importantly, LRx through the amblyopic eye is sufficient to trigger perisynaptic MMP2/9 activity and reduce ECM integrity, suggesting that deprived eye stimulation contributes to the reactivation of plasticity in amblyopic V1. The homeostatic regulation of the threshold for activity-dependent activation of MMP2/9 at thalamo-cortical synapses allows recruitment of this pathway by vision compromised by amblyopia.

## Materials and methods

### Subjects

C57BL/6J mice were purchased from Jackson Laboratory (Bar Harbor, ME). Equal numbers of adult (>postnatal day 90,>P90) males and females were used. Mice were raised in 12 hr light/dark cycle unless specified. Experiments were performed (or subjects were sacrificed) 6 hr into the light phase of a 12:12 hr light:dark cycle. All procedures conformed to the guidelines of the University of Maryland Institutional Animal Care and Use Committee.

### Chronic monocular deprivation

Chronic monocular deprivation was performed at eye opening (P14). Subjects were anesthetized with 2.5% isoflurane in 100% $O_2$ delivered *via* a modified nosecone. The margins of the upper and lower lids of one eye were trimmed and sutured together using a 5–0 suture kit with polyglycolic acid (CP Medical). Subjects were returned to their home cage after recovery at 37°C for 1–2 hr, and disqualified in the event of suture opening.

### MMP2/9 biomarkers and MMP9 inhibitor

MMP2/9 biomarkers (DQ gelatin; D12054, ThermoFisher Scientific, 2 mg/ml, or A580, AS-60554, Anaspec, 10 µg/ml) and MMP9 inhibitor (MMP9 inhibitor I, MMP9i, EMD, 5 nM) were delivered 24 hr prior to LRx through cannulae (2 mm projection, PlasticsOne) implanted ~3 weeks prior to injection. A total volume of 4 µl at 100 nl/min was delivered via a Hamilton syringe attached to a Microsyringe Pump Controller (World Precision Instruments). DQ-gelatin and A580 are exogenous FRET substrates for MMP2/9 in which fluorescence emission is blocked by intramolecular quenching (DQ gelatin; D12054; excitation/emission = 495/519 nm). Proteolysis of the substrate relieves the quenching, such that fluorescence emission reports enzymatic activity. A580 MMP Substrate 1 (AnaSpec) is a 1768 D synthetic peptide ((QXL 570 - KPLA - Nva - Dap(5 - TAMRA) - AR - NH2), containing a C-terminal fluorescent donor carboxy-tetramethyl-rhodamine (TAMRA; 5-TAMRA (547/574 nm Abs/Em) and an N-terminal non-fluorescent acceptor QXL (quencher; Abs 570 nm) for intramolecular FRET. When the molecule is intact, intramolecular FRET (from TAMRA to QXL) quenches fluorescence emission following excitation at 547 nm. However, when the peptide is cleaved by MMP2/9, intramolecular FRET is interrupted, resulting in fluorescence with Em max = 585 nm at Ex = 545 nm. As FRET from donor to acceptor quenches fluorescence, there is no FRET signal resulting from direct activation of the acceptor molecule.

### Antibodies

The following antibodies/dilutions were used: mouse anti-parvalbumin (PV, Millipore) RRID:AB_2174013, 1:2000; rabbit anti-aggrecan (Agg, Millipore) RRID:AB_90460, 1:500; rabbit anti-MMP9 (Cell Signaling) RRID:AB_2144612, 1:2000; mouse anti-β-actin (Sigma-Aldrich) RRID:AB_476744, 1:2000; guinea pig anti-VGluT2 (Millipore) RRID:AB_1587626, 1:2000; followed by appropriate secondary IgG conjugated to Alexa-488, 546 or 647 (Life Technologies) RRID:AB_2534089, RRID:AB_2534093, RRID:AB_2535805, RRID:AB_2534118, 1:1000.

## Immunohistochemistry

Subjects were anesthetized with 4% isoflurane in $O_2$ and perfused with phosphate buffered saline (PBS) followed by 4% paraformaldehyde (PFA) in PBS. The brain was post-fixed in 4% PFA for 24 hr followed by 30% sucrose for 24 hr, and cryo-protectant solution for 24 hr (0.58 M sucrose, 30% (v/v) ethylene glycol, 3 mM sodium azide, 0.64 M sodium phosphate, pH 7.4). Coronal sections (40 µm) were made on a Leica freezing microtome (Model SM 2000R). Sections were blocked with 4% normal goat serum (NGS) in 1X PBS for 1 hr. Antibodies were presented in blocking solution for 18 hr, followed by appropriate secondary antibodies. ECM was visualized with 5 µg/ml fluorescein wisteria floribunda lectin (WFA, Vector Labs) presented during incubation with secondary antibodies.

## Confocal imaging and analysis

Images were acquired on a Zeiss LSM 710 confocal microscope. The cortical distribution of WFA-FITC, Agg and PV immunoreactivity was examined in a z-stack (3 × 10 µm images) acquired with a 10X lens (Zeiss Plan-neofluar 10x/0.30, NA = 0.30) and a z-stack (9–11 × 4.5 µm images) acquired with a 100X (Zeiss Plan-neofluar 100x/1.4 Oil DIC, NA = 1.3). Maximal intensity projections (MIPs; 450 µm width, 0–750 µm from cortical surface) were used to obtain mean intensity profiles in Fiji (NIH). Co-localization of MMP2/9 biomarker puncta with VGluT2 was analyzed in a single Z-section image taken at 40X, using Fiji. After the threshold function (auto threshold + 25) was applied to MMP2/9 biomarker and VGluT2 puncta, co-localized puncta were identified by size exclusion (0.2 $µm^2$ < 2.0 $µm^2$) using the 'analyze particles' function in Fiji. $PV^+$ somata were identified by size exclusion (20–200 $µm^2$) and fluorescence intensity (auto threshold + 25). Co-localization with VGluT2 was re-quantified following 2 µm shift of MMP2/9 biomarker images.

## Fluorescence spectrum measurement of MMP2/9 biomarker

The small peptide MMP2/9 biomarker (A580) was dissolved in PBS (200 ng/ml) and incubated with 100 ng of activated rat recombinant MMP9 (rrMMP9, R and D Systems, according to activation protocol provided by the supplier) at 37° overnight with light protection. Fluorescence spectrum was measured with Cary Eclipse Spectrophotometer (Agilent Technologies).

## Virus injection and cranial window implantation

For two-photon imaging in awake mice, GFP was expressed in excitatory neurons in V1b (AP: 1.0 mm, MD: −3.0 mm, DV: 0.3 mm) following AAV-CaMKII-GFP (titer, 4.3 × 1012 U/ml, UNC Vector Core) injected *via* a Hamilton syringe attached to a Microsyringe Pump Controller (World Precision Instruments) at a rate of 100 nl/min, total volume of 30 nl.

A cranial window consisting of 2, 3 mm diameter coverslips glued with optical adhesive (Norland71, Edmund Optics) to a 5 mm diameter coverslip was implanted as described (*Goldey et al., 2014*). Cannulae (2 mm projection, PlasticsOne) were implanted lateral to the imaging window. The gap between the skull and glass was sealed with silicone elastomer (Kwik-Sil). Instant adhesive Loktite 454 (Henkel) was used to adhere an aluminum head post to the skull and to cover the exposed skull. Black dental cement (iron oxide power, AlphaChemical mixed with white powder, Dentsply) was used to coat the surface to minimize light reflections. Subjects were imaged after 2–3 weeks of recovery.

## Two-photon imaging

Awake subjects were clamped into a holding tube *via* a head post and placed in the dark imaging chamber. Prior to imaging session, the subjects were placed in the holding tube at least twice for 30 min each time for habituation. A two-photon microscope (ThorLabs) controlled by ThorImageLS software with a 16x NA 0.8 water immersion objective lens (Nikon) was used to acquire time lapse fluorescence images. A Chameleon Vision Ti:Sapphire laser (Coherent) was tuned to 940 nm, the excitation wavelength conventionally used for dual imaging (Supplementary materials for deprivation in the visual cortex in *Rose et al., 2016*), in order to simultaneously excite GFP and A580 (Ex max for GFP = 490 nm, for A580 = 547 nm). Fluorescence emission was separated into two channels using a dichroic mirror (cut off at 562 nm) and directed to separate GaAsP photomultiplier tubes (Hamamatsu) to capture GFP and MMP2/9 biomarker signals. Emission filters were placed in front of the PMTs (525 ± 25 nm for green,>568 nm for red). The field of view was 186.5 µm x 186.5 µm (512

$\times$ 512 pixels), 150 to 250 µm from the brain surface. Minimum laser power (59 mW) and PMT gain (60 for green, 100 for red) necessary to image at this depth was used. We confirmed no photobleaching by imaging. With this setting, the bleedthrough from GFP to the red channel was negligible. The same laser power and gain was used for all experiments. Images were acquired at 30 Hz by bidirectional scanning. After control images were acquired in the absence of light stimulation, visual stimulation was delivered at 300 cd/m$^2$ or 150,000 cd/m$^2$ at 1 Hz by a 470 nm LED placed 8 cm from subject's eyes inside the dark imaging chamber. Luminance measurement was performed with Luminance meter nt-1˚ (Minolta). Movement artifacts were corrected with TurboReg plugin in Fiji using the same settings for all experiments with the average intensity of full image stack of GFP fluorescence was used as a template. Mean intensities of MMP2/9 biomarker puncta (30 frames) were analyzed with Fiji in circular regions with no background subtraction. Baseline F for $\Delta$F/F was fluorescence at t = 0.

## Western blot analysis

Mice were anesthetized with isoflurane (4% in 100% O$_2$) and sacrificed following decapitation. The primary visual cortex was rapidly dissected in ice-cold dissection buffer (212.7 mM sucrose, 2.6 mM KCl, 1.23 mM NaH$_2$PO$_4$, 26 mM NaHCO$_3$, 10 mM dextrose, 1.0 mM MgCl$_2$, 0.5 mM CaCl$_2$, 100 µM kynurenic acid; saturated with 95% O$_2$/5% CO$_2$). V1b was isolated using the lateral ventricle and dorsal hippocampal commissure as landmarks. Tissue was homogenized using a Sonic Dismembrator (Model 100, Fisher Scientific) in ice-cold lysis buffer (150 mM NaCl, 1% Nonidet P-40, 50 mM Tris-HCl, pH8.0) containing a protease inhibitor cocktail (Cat#11697498001, Roche). Protein concentration of the homogenate was determined using the BCA Protein Assay kit (Pierce). Equal amounts of total protein (20 µg per lane) were applied to a 12% SDS-polyacrylamide gel for electrophoresis and transferred to a nitrocellulose membrane. The membranes were incubated with a blocking solution (4% skim milk in 1X PBS) for 30 min. Primary antibodies were presented in the blocking solution for 2 hr, followed by appropriate secondary antibodies. Immunoreactive bands were visualized with a Typhoon TRIO Variable Imager (GE Healthcare). The intensity of immunoreactive bands of active MMP9 (95 kDa), which can be distinguished from inactive pro-MMP9 (~105 kDa, [*Szklarczyk et al., 2002*]), were analyzed using ImageQuant TL (rubber band background subtraction; GE Healthcare), and normalized to β-actin (38 kDa).

## Acknowledgements

We thank Andrew Borrell for illustration in *Figure 5A*.

## Additional information

### Funding

| Funder | Grant reference number | Author |
|--------|------------------------|--------|
| National Eye Institute | R01EY016431 | Elizabeth M Quinlan |
| National Institute on Deafness and Other Communication Disorders | R01DC009607 | Patrick O Kanold |

The funders had no role in study design, data collection and interpretation, or the decision to submit the work for publication.

### Author contributions

Sachiko Murase, Dan Winkowski, Data curation, Formal analysis, Investigation, Methodology; Ji Liu, Data curation, Formal analysis, Funding acquisition, Investigation, Methodology; Patrick O Kanold, Elizabeth M Quinlan, Conceptualization, Data curation, Formal analysis, Funding acquisition, Investigation, Methodology, Project administration

## Author ORCIDs

Sachiko Murase (iD) http://orcid.org/0000-0002-9078-0471
Patrick O Kanold (iD) http://orcid.org/0000-0002-7529-5435
Elizabeth M Quinlan (iD) https://orcid.org/0000-0003-3496-6607

## Ethics

Animal experimentation: All procedures, under Quinlan lab protocol R-MAY-18-25, conformed to the guidelines of the University of Maryland Institutional Animal Care and Use Committee and the Guide for the Care and Use of Laboratory Animals of the National Institutes of Health.

## Decision letter and Author response

Decision letter https://doi.org/10.7554/eLife.52503.sa1
Author response https://doi.org/10.7554/eLife.52503.sa2

# Additional files

## Supplementary files

• Transparent reporting form

## Data availability

All data generated/analysed during this study are included in the manuscript and supporting files. Source data files have been provided for Figures 1-5.

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
