## [Decision Letter]

**Acceptance summary:**

Prolonged loss of visual input through one eye decreases the strength and selectivity of neuronal responses to that eye in the visual cortex, a process called amblyopia. However, subsequent reduction of vision in both eyes (dark rearing) can restore plasticity, permitting recovery of function. Previously, the authors showed that recovery of function requires activation of a metalloproteinase. The authors add to their earlier study by showing that the visual experience required to restore cortical plasticity by activating this metalloproteinase can occur through the amblyopic eye. This has important implications for clinical treatment of this disorder. In addition, the authors demonstrate a novel FRET assay for perisynaptic proteolysis in vivo.

**Decision letter after peer review:**

Thank you for submitting your work entitled "Homeostatic regulation of perisynaptic MMP9 activity in the amblyopic visual cortex" for consideration by *eLife*. Your article has been reviewed by two peer reviewers, and the evaluation has been overseen by a Reviewing Editor and a Senior Editor. The reviewers have opted to remain anonymous.

It is potentially suitable as a Research Advance following your previous study, but only if the claims of the manuscript can be further substantiated. In this case, the core observations need additional clarification and documentation, and probably additional experiments. In the discussion between the reviewers and editors, there was overall enthusiasm for the paper and so we encourage a resubmission, provided the core issues can be addressed. In addition to providing edited versions of the full reviews below, we have tried to indicate the strengths of the study and the key issues that need to be addressed in the revision.

Overall strengths of the study:

The authors add to their earlier study by showing that the visual experience required to restore cortical plasticity by activating matrix metaloprotease activity can occur through the amblyopic eye. This has important implications for clinical treatment of this disorder. In addition, the authors attempt to demonstrate a novel FRET assay for perisynaptic proteolysis in vivo. This latter portion of the study potentially increases its impact, but is not compelling in its present form.

Overall revision plan:

The key clarifications or additional experiments needed involve:

1) measurements that calibrate the 2-photon FRET imaging and that convincingly demonstrate a significant increase (e.g. the time course of that increase) after light exposure. Calibration (cited or done themselves) is needed to give us confidence that the sensor actually detects what it claims (other than the words on the manufacturer's website). It seems (from checking the manufacture's website) that they are activating the FRET by stimulating the GFP but it is not stated. The following questions need an answer: How was the DF/F calculated relative to the GFP signal, or biomarker signal? What evidence do we have that the imaging conditions were similar across the conditions?

2) More rigorous analysis of the co-localization including estimates that colocalization would be obtained by chance.

*Reviewer #1:*

The article by Murase et al. describes that one of the molecular mechanisms that is induced by light recovery following dark exposure (activation of MMP9), occurs in amblyopic animals just as it does in typically-raised animals (shown by Murase et al., 2017). In addition, the article provides evidence that expression of MMP9 occurs after dark exposure followed by light exposure at medium intensities (300 cd/m^2; matching a typical LCD computer monitor) or bright intensities (150k cd/m^2, about daylight) after either dark exposure or a shorter-duration (18 hours) dark exposure.

There are several methodological problems that will need correction.

First, the colocalization analysis is not rigorous. In order for the colocalization rates to be accurately reported, one would need to establish the expected rate of empirically counting a colocalization when there is not a synapse. In cortico-cortical synapses and in retina, confocal techniques are hopeless for this, as the density of synapses is higher than the resolution of the technique. It is possible that this is acceptable given the density of thalamocortical synapses, but no analysis is given that says this is the case. A relative increase can be argued based on the data here, but it is not *eLife*-quality in my opinion.

Second, the 2-photon imaging is poorly documented and poorly described. What is the FRET calibration? Apparently we are to take some δ F / F as evidence of some signal. Showing the signal over time (maybe measured every 1-2 minutes) would be one way to show convincingly that the signal increases in a meaningful way. Showing that the green signal is constant would also be helpful, but it would be better to quantify the FRET. Is there a citation where this is all given that is missing?

Third, it is important to give some comparisons for the light source. The luminance is described in units appropriate for a point source, but a surface is shown in Figure 5A. I imagined that 300 cd/m^2 is similar to a computer monitor in a dark room, and that 150k cd/m^2 is like daylight. This begs the question as to why 18 hours of darkness followed by typical daylight exposure might not be useful therapeutically? Perhaps this is an interesting discovery to be explored further?

Fourth, like 5% of scientists, I am red-green color-blind and can hardly make heads or tails of the red/green figures. It would be better to replace red with magenta (magenta = red + blue).

*Reviewer #2:*

In this manuscript by Murase et al., the authors expand on their previous finding outlining a case for a role of matrix metalloproteinases in restoring visual cortical plasticity with light reintroduction after dark exposure published in *eLife* in 2017. In the current study, the authors present compelling evidence that some of the same mechanisms (activation of MMP9) can be induced in a mouse model of amblyopia. The effect appears to be a robust one, with impressive differences in staining for their biomarker of MMP activity and for various extracellular matrix components throughout the cortical depth upon light reexposure. This Research Advance has important translational relevance in that it provides pre-clinical evidence that such a strategy could be promising as a way of restoring normal visions to patients with amblyopia. The data support the conclusions drawn. As the experiments were performed in a rigorous way, with methods similar to what the authors have published previously in *eLife*, I have only a few suggestions and requests for clarification [minor comments not shown].

[Editors’ note: what now follows is the decision letter after the authors submitted for further consideration.]

Thank you for resubmitting your work entitled "Homeostatic regulation of perisynaptic MMP9 activity in the amblyopic visual cortex" for further consideration by *eLife*. Your revised article has been evaluated by Andrew King as the Senior Editor, Sacha Nelson as the Reviewing Editor, and two peer reviewers.

The manuscript has been improved but there are some remaining issues that need to be addressed before acceptance, as outlined below:

The reviewers and editors agreed that the additional control for the analysis of co-localization (outlined below) is straightforward to implement, could be done quickly and would strengthen the paper.

Reviewer #1:

The manuscript is much improved and addresses most of the concerns very well.

I am a bit concerned that the 90 degree rotation as a control for the co-localization analysis doesn't really address the issue of chance co-localization. There are some regions of the image that correspond to cell bodies, which are practically without any label, so rotating the synaptic label onto these regions will naturally result in lower co-localization. It would be better to shift one of the images by 2 μm (or whatever the maximum exclusion zone size is) and re-calculate the co-localization to get a better sense of the chance co-localization in the regions where label is abundant.

---

## [Author Response]

[Editors’ note: the author responses to the first round of peer review follow.]

Overall revision plan:The key clarifications or additional experiments needed involve:1) measurements that calibrate the 2-photon FRET imaging and that convincingly demonstrate a significant increase (e.g. the time course of that increase) after light exposure. Calibration (cited or done themselves) is needed to give us confidence that the sensor actually detects what it claims (other than the words on the manufacturer's website). It seems (from checking the manufacture's website) that they are activating the FRET by stimulating the GFP but it is not stated.

Thank you for the opportunity to clarify: the biomarker employed for 2P live imaging A580 MMP Substrate 1 (AnaSpec) is a 1768 D synthetic peptide substrate for MMP 2/9 ((QXL 570 – KPLA – Nva – Dap(5 – TAMRA) – AR – NH2), containing a C-terminal fluorescent donor carboxy-tetramethyl-rhodamine (TAMRA; 5-TAMRA (547/574 nm Abs/Em) and an N-terminal non-fluorescent acceptor QXL (quencher; Abs 570 nm) for intramolecular FRET. When the molecule is intact, i.e. in the absence of MMP 2/9 activity, intramolecular FRET (from TAMRA to QXL) quenches fluorescence emission following excitation at 547 nm. However, when the peptide is cleaved by *MMP2*/9, intramolecular FRET is interrupted, resulting in fluorescence with Em max = 585 nm at Ex = 545 nm (Figure 5B). In FRET applications that involve transfer from a fluorescent donor to a fluorescent acceptor, calibration is necessary to measure the fluorescence emission of the acceptor in the absence of FRET (i.e. direct activation of the acceptor during excitation of the donor), as a baseline to which all FRET signals are calibrated. However, the MMP biomarker peptide employed here, the FRET acceptor is non-fluorescent, such that FRET from donor to acceptor quenches fluorescence. In this case there is no FRET signal resulting from direct activation of the acceptor molecule, deeming calibration unnecessary. This is now included in the Materials and methods (subsection “MMP2/9 biomarkers and MMP9 inhibitor”).

In new Figure 5B, we present the fluorescent emission spectrum of the MMP 2/9 biomarker cleaved in vitro by recombinant active MMP9 with Em max = 585 nm at Ex = 545 nm. As described in the Materials and methods, in our 2P live imaging, the laser was tuned to 940 nm, the excitation wavelength conventionally used for dual acquisition of GCaMP and mRuby (Ex max for GCaMP=490 nm, for mRuby=558 nm, (Rose et al., 2016)), in order to simultaneously excite GFP and A580 (Ex max for GFP=490 nm, for A580=547 nm). The emission was separated into two channels using a dichroic mirror (cut off at 562 nm). In our experiments, we imaged 150 – 250 µm from the brain surface, using minimum laser power (59 mW) and PMT gain (60 for green, 100 for red). With this setting, the bleed-through from green to the red channel was negligible. This is now included in the Materials and methods (subsection “Two-photon imaging”).

In response to the second comment, we have performed new experiments that quantify the time-course of the light-induced increase in MMP biomarker fluorescence. Specifically, we show in the absence of light stimulation (0 cd/m^2^) in a single DE subject raw fluorescent intensity (pixels) and DF/F are unchanged for 50 seconds. Similarly, there is no change in the simultaneously acquired GFP signal over this time course. In contrast, light stimulation (300 cd/m^2^) induces a ~ 40% increase in raw fluorescent intensity (pixels) and DF/F of the biomarker from -10 seconds to +40 seconds of visual stimulation, but no change in simultaneously acquired GFP (new Figure 5C). Population data (12 puncta from 3 subjects; new Figure 5D) shows that the increase in biomarker DF/F induced by 300 cd/m^2^ light stimulation is significantly differently from no light control (0 cd/m^2^), repeated measures ANOVA, F_(1, 22)_, *p<0.001). In addition, we demonstrate the specificity of the biomarker for reporting MMP9 activity in new Figure 5E, in which the increase in biomarker DF/F induced by visual stimulation at 150,000 cd/m^2^ is blocked by pre-treatment with a potent and specific inhibitor of MMP9 (MMP9 inhibitor I, CAS 1177749-58-4; IC50=5 nM); 5 nM 4 µl at 100 nl/min) p=0.13 Student’s T-test. This is included in the Results (subsection “DE regulates threshold for perisynaptic MMP2/9 activation”).

The following questions need an answer: How was the DF/F calculated relative to the GFP signal, or biomarker signal? What evidence do we have that the imaging conditions were similar across the conditions?

DF/F of biomarker is reported as raw intensity (pixels) and DF/F, and is not calculated relative to GFP. As described in the Materials and methods, in our 2P live imaging, the laser was tuned to 940 nm, the excitation wavelength conventionally used for dual acquisition of GCaMP and mRuby (Exmax for GCaMP=490 nm, for mRuby=558 nm, (Rose et al., 2016)), to simultaneously excite GFP and A580 (Exmax for GFP=490 nm, for A580=547 nm). The emission was separated into two channels using a dichroic mirror (cut off at 562 nm). New Figure 5C shows that visual stimulation after DE induces an increase in biomarker DF/F, while the simultaneously acquired GFP DF/F is stable. This is included in the Results (subsection “DE regulates threshold for perisynaptic MMP2/9 activation”).

In our experiments, we imaged 150 – 250 µm from the brain surface, using minimum laser power (59 mW) and PMT gain (60 for green, 100 for red). With this setting, the bleed-through from green to the red channel was negligible. This is now included in the Materials and methods (subsection “Two-photon imaging”).

2) More rigorous analysis of the co-localization including estimates that colocalization would be obtained by chance.

To control for false positive co-localization of VGluT2 and MMP biomarker, we re-quantified colocalization after 90-degree rotation of the biomarker image. Following rotation, co-localization was very low, and in each case the co-localization was not regulated by visual history: cMD dep: 4.8%, cMD nondep:3.4%, LRx dep: 7.2%, LRx non-dep: 7.5%, added to Figure 1B and dep:9.2 ± 3.5%; non: 4.2 ± 0.8% in new Figure 3B (paired Student’s T-test vs. images in correct register, all p<0.01).

Reviewer #1:The article by Murase et al. describes that one of the molecular mechanisms that is induced by light recovery following dark exposure (activation of MMP9), occurs in amblyopic animals just as it does in typically-raised animals (shown by Murase et al., 2017). In addition, the article provides evidence that expression of MMP9 occurs after dark exposure followed by light exposure at medium intensities (300 cd/m^2; matching a typical LCD computer monitor) or bright intensities (150k cd/m^2, about daylight) after either dark exposure or a shorter-duration (18 hours) dark exposure.There are several methodological problems that will need correction.First, the colocalization analysis is not rigorous. In order for the colocalization rates to be accurately reported, one would need to establish the expected rate of empirically counting a colocalization when there is not a synapse. In cortico-cortical synapses and in retina, confocal techniques are hopeless for this, as the density of synapses is higher than the resolution of the technique. It is possible that this is acceptable given the density of thalamocortical synapses, but no analysis is given that says this is the case. A relative increase can be argued based on the data here, but it is not eLife-quality in my opinion.

To control for false positive co-localization of VGluT2 and MMP biomarker, we re-quantified colocalization after 90-degree rotation of the biomarker image. Following rotation, co-localization was very low, and in each case the co-localization was not regulated by visual history: cMD dep: 4.8%, cMD nondep:3.4%, LRx dep: 7.2%, LRx non-dep: 7.5%, added to Figure 1B and dep:9.2 ± 3.5%; non: 4.2 ± 0.8% in new Figure 3B (paired Student’s T-test vs. images in correct register, all p<0.01).

Second, the 2-photon imaging is poorly documented and poorly described. What is the FRET calibration? Apparently we are to take some δ F / F as evidence of some signal. Showing the signal over time (maybe measured every 1-2 minutes) would be one way to show convincingly that the signal increases in a meaningful way. Showing that the green signal is constant would also be helpful, but it would be better to quantify the FRET. Is there a citation where this is all given that is missing?

The biomarker employed for 2P live imaging A580 MMP Substrate 1 (AnaSpec) is a 1768 D synthetic peptide substrate for MMP 2/9 ((QXL 570 – KPLA – Nva – Dap(5 – TAMRA) – AR – NH2), containing a Cterminal fluorescent donor carboxy-tetramethyl-rhodamine (TAMRA; 5-TAMRA (547/574 nm Abs/Em) and an N-terminal non-fluorescent acceptor QXL (quencher; Abs 570 nm) for intramolecular FRET. When the molecule is intact, i.e. in the absence of MMP 2/9 activity, intramolecular FRET (from TAMRA to QXL) quenches fluorescence emission following excitation at 547 nm. However, when the peptide is cleaved by *MMP2*/9, intramolecular FRET is interrupted, resulting in fluorescence with Em max = 585 nm at Ex = 545 nm (Figure 5B). In FRET applications that involve transfer from a fluorescent donor to a fluorescent acceptor, calibration is necessary to measure the fluorescence emission of the acceptor in the absence of FRET (i.e. direct activation of the acceptor during excitation of the donor), as a baseline to which all FRET signals are calibrated. However, the MMP biomarker peptide employed here, the FRET acceptor is nonfluorescent, such that FRET from donor to acceptor quenches fluorescence. In this case there is no FRET signal resulting from direct activation of the acceptor molecule, deeming calibration unnecessary. This is now included in the Materials and methods (subsection “MMP2/9 biomarkers and MMP9 inhibitor”).

In new Figure 5B, we present the fluorescent emission spectrum of the MMP 2/9 biomarker cleaved in vitro by recombinant active MMP9 with Em max = 585 nm at Ex = 545 nm. As described in the Materials and methods, in our 2P live imaging, the laser was tuned to 940 nm, the excitation wavelength conventionally used for dual acquisition of GCaMP and mRuby (Ex max for GCaMP=490 nm, for mRuby=558 nm, (Rose et al., 2016)), in order to simultaneously excite GFP and A580 (Ex max for GFP=490 nm, for A580=547 nm). The emission was separated into two channels using a dichroic mirror (cut off at 562 nm). In our experiments, we imaged 150 – 250 µm from the brain surface, using minimum laser power (59 mW) and PMT gain (60 for green, 100 for red). With this setting, the bleed-through from green to the red channel was negligible. This is now included in the Materials and methods (subsection “Two-photon imaging”).

In response to the second comment, we have performed new experiments that quantify the time-course of the light-induced increase in MMP biomarker fluorescence. Specifically, we show in the absence of light stimulation (0 cd/m^2^) in a single DE subject raw fluorescent intensity (pixels) and DF/F are unchanged for 50 seconds. Similarly, there is no change in the simultaneously acquired GFP signal over this time course. In contrast, light stimulation (300 cd/m^2^) induces a ~ 40% increase in raw fluorescent intensity (pixels) and DF/F of the biomarker from -10 seconds to +40 seconds of visual stimulation, but no change in simultaneously acquired GFP (new Figure 5C). Population data (12 puncta from 3 subjects; new Figure 5D) shows that the increase in biomarker DF/F induced by 300 cd/m^2^ light stimulation is significantly differently from no light control (0 cd/m^2^), repeated measures ANOVA, F_(1, 22)_, *p<0.001). In addition, we demonstrate the specificity of the biomarker for reporting MMP9 activity in new Figure 5E, in which the increase in biomarker DF/F induced by visual stimulation at 150,000 cd/m^2^ is blocked by pre-treatment with a potent and specific inhibitor of MMP9 (MMP9 inhibitor I, CAS 1177749-58-4; IC50=5 nM); 5 nM 4 µl at 100 nl/min) p=0.13 Student’s T-test. This is included in the Results (subsection “DE regulates threshold for perisynaptic MMP2/9 activation”).

Third, it is important to give some comparisons for the light source. The luminance is described in units appropriate for a point source, but a surface is shown in Figure 5A. I imagined that 300 cd/m^2 is similar to a computer monitor in a dark room, and that 150k cd/m^2 is like daylight. This begs the question as to why 18 hours of darkness followed by typical daylight exposure might not be useful therapeutically? Perhaps this is an interesting discovery to be explored further?

Thank you, we have corrected this error and changed the illustration to a point light source.

The description of luminance measurement was expanded in the Materials and methods:

“Luminance measurement was performed with Luminance meter nt-1° (Minolta).”

Reference to environmental luminance was described in the Results (subsection “DE regulates threshold for perisynaptic MMP2/9 activation”):

– “moderate intensity light (300 cd/m^2^, luminance ~ luminance in the laboratory room)”;

– “high intensity light (150,000 cd/m^2^, brightness equivalent to direct sunlight at noon)”.

As for the response of dark-adopted (18 hr) subjects to high intensity stimulation, we incorporated reviewer’s suggestion in the Discussion:

“Indeed, following 10 days of DE, ambient light was sufficient to drive a significant increase in perisynaptic MMP 2/9. LRx induced increase in biomarker fluorescence was blocked by a potent and specific inhibitor of MMP9. Notably, 18 hours of dark adaptation induced response to high intensity light.”

Fourth, like 5% of scientists, I am red-green color-blind and can hardly make heads or tails of the red/green figures. It would be better to replace red with magenta (magenta = red + blue).

Apologies, we have changed the color scheme throughout.

[Editors' note: the author responses to the re-review follow.]

The manuscript has been improved but there are some remaining issues that need to be addressed before acceptance, as outlined below:The reviewers and editors agreed that the additional control for the analysis of co-localization (outlined below) is straightforward to implement, could be done quickly and would strengthen the paper.Reviewer #1:The manuscript is much improved and addresses most of the concerns very well.I am a bit concerned that the 90 degree rotation as a control for the co-localization analysis doesn't really address the issue of chance co-localization. There are some regions of the image that correspond to cell bodies, which are practically without any label, so rotating the synaptic label onto these regions will naturally result in lower co-localization. It would be better to shift one of the images by 2 μm (or whatever the maximum exclusion zone size is) and re-calculate the co-localization to get a better sense of the chance co-localization in the regions where label is abundant.

We have performed this additional control for the analyses of co-localization of the MMP biomarker puncta and puncta for thalamic axons revealed by VGlut2. Figures 1 and 3 contain new analyses confirming that a 2 μm shift of the biomarker image reduces co-localization with VGlut2 and abolishes changes in co-localization induced our experimental manipulation (light reintroduction):

– Figure 1: “To control for false positive co-localization, we analyzed the co-localization following a 2 μm shift of the biomarker image relative to VGlut2. Following this manipulation we observe low co-localization of the two fluorescent signals (cMD dep: 7.8 ± 3.0%, cMD non-dep: 5.3 ± 2.7%, LRx dep: 6.9 ± 3.0%, LRx non-dep: 8.4 ± 3.0%) which differ significantly from co-localization observed with the correct registration of VGluT2 and MMP biomarker images (cMD dep: p=0.0029, cMD non-dep: p=0.036, LRx dep: p=1.5 x 10^-5^, LRx non-dep: p=9.5 x 10^-6^, paired Student’s T-Test, Figure 1B).”

– Figure 3: “Co-localization with VGluT2 following a 2 μm shift of the biomarker image was low: dep:8.0 ± 3.9%; non: 7.0 ± 3.9%, and significantly different from co-localization when the two images were correctly registered (dep: p=3.2 x 10^-5^; non: p=0.0017, paired Student’s T-test; Figure 3B).”